# Extant Earthly Microbial Mats and Microbialites as Models for Exploration of Life in Extraterrestrial Mat Worlds

**DOI:** 10.3390/life11090883

**Published:** 2021-08-27

**Authors:** Bopaiah Biddanda, Anthony Weinke, Ian Stone, Scott Kendall, Phil Hartmeyer, Wayne Lusardi, Stephanie Gandulla, John Bright, Steven Ruberg

**Affiliations:** 1Annis Water Resources Institute, Grand Valley State University, Muskegon, MI 49441, USA; weinkea@mail.gvsu.edu (A.W.); stoneia@mail.gvsu.edu (I.S.); 2Biology Department, Muskegon Community College, Muskegon, MI 49442, USA; scott.kendall@muskegoncc.edu; 3Thunder Bay National Marine Sanctuary, Office of National Marine Sanctuaries, National Oceanic and Atmospheric Administration, Alpena, MI 49707, USA; phil.hartmeyer@noaa.gov (P.H.); wayne.lusardi@noaa.gov (W.L.); steph.gandulla@noaa.gov (S.G.); john.bright@noaa.gov (J.B.); 4Great Lakes Environmental Research Laboratory, National Oceanic and Atmospheric Administration, Ann Arbor, MI 48108, USA; steve.ruberg@noaa.gov

**Keywords:** mat worlds, microbial mats, microbialites, cyanobacteria, motile microbes, photosynthesis, chemosynthesis, extraterrestrial life models, exobiology

## Abstract

As we expand the search for life beyond Earth, a water-dominated planet, we turn our eyes to other aquatic worlds. Microbial life found in Earth’s many extreme habitats are considered useful analogs to life forms we are likely to find in extraterrestrial bodies of water. Modern-day benthic microbial mats inhabiting the low-oxygen, high-sulfur submerged sinkholes of temperate Lake Huron (Michigan, USA) and microbialites inhabiting the shallow, high-carbonate waters of subtropical Laguna Bacalar (Yucatan Peninsula, Mexico) serve as potential working models for exploration of extraterrestrial life. In Lake Huron, delicate mats comprising motile filaments of purple-pigmented cyanobacteria capable of oxygenic and anoxygenic photosynthesis and pigment-free chemosynthetic sulfur-oxidizing bacteria lie atop soft, organic-rich sediments. In Laguna Bacalar, lithification by cyanobacteria forms massive carbonate reef structures along the shoreline. Herein, we document studies of these two distinct earthly microbial mat ecosystems and ponder how similar or modified methods of study (e.g., robotics) would be applicable to prospective mat worlds in other planets and their moons (e.g., subsurface Mars and under-ice oceans of Europa). Further studies of modern-day microbial mat and microbialite ecosystems can add to the knowledge of Earth’s biodiversity and guide the search for life in extraterrestrial hydrospheres.

## 1. Introduction and Background

Whenever we look up at the night sky and wonder if there is life out there somewhere, our minds conjure images of water worlds, because life, as we know it, is water-based. Earth is a water-dominated planet located in the habitable, or “goldilocks”, zone around the Sun. Indeed, the bulk of Earth’s biosphere is composed of water, wherein the majority of life is microbial—both in terms of numbers and biomass of organisms. Microbes less than a tenth of a millimeter (<100 µm) in size occur in counts of ~1 million cells per mL in natural waters ranging from a backyard puddle to the deep ocean [1]. Thus, life in our planet’s hydrosphere is ultimately an interconnected global ecosystem run by microbes [2,3]. Arguably, a more intimate understanding of everyday microbes in the soil, water, and air on Earth would advance the search for possible life in extraterrestrial water worlds. Extant microbial mat worlds already present on Earth may provide useful models and practical analogs in the search for life on other planets and their moons in the solar system and beyond.

Evidence of the widespread distribution of microbial mats during the Archean and Proterozoic eons is found today in the geological record of microbialites, such as stromatolites [4]. These modern-day mat ecosystems resemble life during the Precambrian era, when the biosphere was mostly microbial and benthic [5,6,7,8]. Fortunately, even today, microbial mats similar to life in the shallow, anoxic, sometimes euxinic (high hydrogen sulfide, low dissolved oxygen) seas of the early biosphere thrive in several globally distributed refugia under extreme conditions of moisture, salinity, pH, temperature, and oxygen [9,10,11,12,13,14,15,16,17]. Indeed, a great variety of living microbial mats with a diversity of structural and functional characteristics are found in extant extreme ecosystems all over the world ranging from subglacial lakes to deep sea thermal vents [14]. These extremophile microbial mats and their exceptional environments provide readily accessible and working scenarios that could assist us in better envisioning and exploring for life elsewhere in our solar system, such as in Mars, Enceladus, and Titan (Saturnian moons), and Ganymede and Europa (Jovian moons) [18].

Herein, we describe life in two extant, microbially driven, subsurface mat ecosystems characterized by two key microbial mat types in Earth’s waters and discuss their relevance for exploration of mat worlds that may exist in the solar system and beyond. We do this by first introducing the two mat ecosystems, which are found in the submerged sinkholes of Lake Huron (Michigan, USA) and in the shallow coastal waters of Laguna Bacalar (Yucatan Peninsula, Mexico):Extensive soft and colorful microbial mats on the sediment surface of submerged sinkholes (~1–200 m) in Lake Huron, a freshwater Laurentian Great Lake (USA) (45.198790°, −83.327769°) (Appendix A).Semi-hard giant microbialite reefs in the shallow coastal waters (~0–5 m) of Laguna Bacalar, Quintana Roo, Mexico (18.644518°, −88.405611°) (Appendix A).

We follow-up with a discussion regarding relevant aspects of how we have studied these ecosystems over the past two decades and scenarios of how these and additional approaches may be useful in studies of potential mat worlds encountered in other extraterrestrial bodies in the solar system.

## 2. Microbial Mats in Lake Huron’s Submerged Sinkholes

Sulfur-rich environments inhabited by cyanobacteria, akin to those that were present long ago in Earth’s history, are found in the coastal waters of NW Lake Huron—a region underlain by karstic limestone wherein groundwater seeps through Paleozoic marine evaporites (Figure 1 and Figure 2, Appendix A) [13]. Here, dissolution of Silurian-Devonian bedrock has produced karstic sinkhole formations on the lake floor, through which anoxic, sulfurous groundwater continually vents onto the lake floor, forming a chemocline with the overlying lake water and fueling the growth of microbial mats. The venting groundwater is characterized by nearly constant temperature (9.5 °C), pH (7.3) and oxygen (0–2 mg L^−1^) and relatively high chloride (25 mg L^−1^), sulfate (>1000 mg L^−1^), and conductivity (2300 µS cm^−1^) compared with overlying Lake Huron water [19]. The denser ground water hugs the bottom of the sinkhole as it flows out, providing a stable hypoxic and sulfur-rich habitat for microbes. Among the many unknown factors related to these sinkholes and the microbes that inhabit them are what is the age of the ground water, how microbes came to colonize these sinkholes, and how they disperse to other sinkholes.

Microbial mats that cover the floor and walls of these shallow submerged sinkholes are primarily comprised of communities of photosynthetic cyanobacteria and chemosynthetic, sulfur-oxidizing bacteria (SOB; Figure 1 and Figure 2) [13,20]. The cyanobacteria are purple-pigmented and of *Planktothrix* (formerly *Oscillatoria*) and *Phormidium* genera, which are capable of oxygenic photosynthesis (OP) and anoxygenic photosynthesis (AP), while the SOB are white-appearing and hail from the genus *Beggiatoa* (Figure 3) [13,20,21]. Directly beneath the mats is a layer of white carbonate granules. However, the carbonate layer does not build up and, consequently, these mats are not reef builders. Presumably, the day’s worth of carbonate crystals produced during photosynthesis is redissolved each night during respiration. While we have no direct proof of this mechanism describing why the carbonate layer does not accumulate over time, microprobe measurements of the mat-sediment complex do suggest large fluctuations in pH between day and night—higher in the day and lower at night—that may lead to night-time dissolution of the carbonate layer laid down during the day ([21], unpublished data). Successive layers of sulfate-reducing bacteria (SRB) thrive in the organic matter-rich sediments beneath these mats and, together with the producer communities at the top, establish a functioning redox tower [13,22]. These mat ecosystems are closely related and share overall mat morphological similarity to those inhabiting the bottom of terrestrial hot springs distributed worldwide and ice-covered Antarctic lakes [14,23,24], and are analogous to the most ancient forms of life on Earth [25,26]. These features make them model soft-mat systems for exploring the biosphere’s evolution, biodiversity, and physiology under extreme environmental conditions [7,13,27]. It is also likely that microbial mat worlds we may run into in space exploration are analogous to those currently found on Earth. In any case, the diversity of extant mat ecosystems provides the best working models available.

## 3. Massive Microbialites of Laguna Bacalar

The Yucatan Peninsula is a carbonate platform characterized by karstic aquifers, caves, and cenotes (sinkholes) that are widely present throughout the peninsula wherein warm waters contain high calcium carbonate and bicarbonate [28]. Cenotes serve as a source of groundwater and as traps for organic carbon material entering from the surrounding land as well as nurture benthic mats and support microbialites along the shallow edges [29]. This is similar to the sinkholes in Lake Huron, where motile microbial mat filaments climb over falling planktonic debris and permanently bury them in the anoxic sediment below [30]. Laguna Bacalar—called “lake of seven colors” by the Maya for its varying shades of turquoise and blue—is home to one of the largest freshwater microbialite structures in the world, which runs along the shoreline for ~15 km (Figure 4; Appendix A). Huge, cauliflower-like microbialites (living rocks, see below) growing in the oligotrophic Laguna Bacalar are organo-sedimentary calcium carbonate reef structures formed by the secretory and matting behavior of cyanobacteria and associated algae growing in the calcium, sulfate, and bicarbonate-rich karstic waters [31,32,33]. Such hard mat morphology is found globally, such as stromatolites in Shark Bay, Australia, Tufas in Big Soda Lake Nevada, among others, which gives support to using the microbialites of Laguna Bacalar as models for hard mats [14,15,25].

Reef-building, filamentous cyanobacteria inside the microbialites precipitate calcium carbonate during photosynthesis, rising as the reef builds up around them to stay within the sunlit zone. In the Laguna Bacalar microbialites, life appears to hide just millimeters beneath the limestone surface, likely in order to prevent UV-photodamage. Recent studies of the Laguna Bacalar microbialite microbiome has shown that it consists primarily of proteobacteria, cyanobacteria, and bacteroidetes, followed by other groups in smaller proportions, revealing a diverse and complex community [33]. Thus, extremely little life signature may prevail on the surface despite a thriving layer of life just beneath it. The fact that life occurs hidden in subsurface habitats on Earth (either below a hard surface, in crevices, or underwater) makes the case for why we should be probing similar subsurface zones in our search for extraterrestrial life.

In Laguna Bacalar, Mexico, the microbialite structures on the meter scale appear to be minerals (Figure 4), though the unusual roundish pod-like formation suggests a biogenic origin. Having a firm consistency (unlike the Lake Huron mats), a hand knife was adequate to cut cross sections of the microbialites for observation. On the centimeter scale, strata indicating different levels of redox are apparent with distinctively pigmented blue-green and purple layers, signifying the living layer (Figure 5A,B). On the micrometer scale, closely packed pigmented filamentous cells are readily apparent even with the crudest imaging methods (Figure 5C,D). Previous microscopic observations [32] and our personal observations (Figure 5C,D) suggest that the subsurface blue-green and purple layers are primarily composed of filamentous cyanobacteria of the genera *Calothrix* and *Scytonema,* respectively—although there is no way of independently confirming their identities without isolation and follow up genetic analysis. While motility of these filaments has not been observed in the field, they are likely capable of both horizontal and vertical movements, as evidenced by their ability to stay in layered configuration just millimeters below the surface of the microbialite, which is ever-growing outwards. Phototactic and chemotactic motility of such mat-building filaments has been chronicled in the microbial mats colonizing the submerged sinkholes of Lake Huron [34,35], which supports the notion that microbes in hard substrate microbialites are similarly motile to those in the soft-substrate microbial mats.

Assuming that extant soft mat and hard microbialite ecosystems on Earth bear resemblance to life forms we may encounter in extraterrestrial habitats, then, in addition to remotely inferring biosignatures from afar, the unambiguous detection of extant prokaryotic life is to directly look for biosignatures, community structure, species composition, biogeochemical context, and metabolic activity—including physical motility [36]. Indeed, remote inferences indicating the possible presence of life from afar (such as reflectance spectra of pigmented microbes) [37] can set the stage for follow-up fly-bys and landings that involve direct, on-site studies. Sampling and analyses likely would include not only the standard search for chemical biosignatures of life, such as biomolecules and isotope ratios [38,39], but also in situ microscopy for imaging microbial cells and tracking their movements across the micrometer scale [40,41]. Indeed, numerous emerging miniaturized life-detection technologies capable of in situ spectroscopy and digital imaging are being prepared to board future space missions in search of bioelements, biomolecules, and microbes—both dead and living.

## 4. How to Study Mat Worlds of Extraterrestrial Hydrospheres When We Get There

Realistic scenarios of technology application for accessing and studying extant mat ecosystems can be useful for carrying out successful operations in extraterrestrial settings. Lake Huron’s submerged sinkholes near Middle Island are easily accessed from the nearby Rockport State Park or Presque Isle State Harbor on the mainland (Appendix A). NOAA operates a national marine sanctuary based in the nearby town of Alpena, MI, which encompass the area of Lake Huron where numerous submerged sinkholes are present, including Middle Island. Middle Island has a dock and lighthouse operating year-round, and a bed and breakfast in operation during the summer season. Thus, Middle Island could constitute a convenient base for dive and remotely operated vehicle/autonomous underwater vehicle (ROV/AUV) operations at the adjacent Middle Island Sinkhole (MIS; Figure 6). At MIS, vibrant purple and white mats that are found at the source of the groundwater gradually give way to less vibrantly colored mats that dwindle away into the usual grey lake bottom at a distance of 100–200 m. This gradient could serve for sequential video and sampling operations to look for changing characteristics of the environment and any corresponding changes in the composition of life—such as species-level changes and attendant shifts in ecosystem processes. For example, neutral density dyes or sterile fluorescently labelled colloids could be injected in flowing environments and followed by ROV/AUVs to record the flow rate and any attendant life.

Time-lapse photographic equipment could also be deployed to monitor any diel changes in visual mat characteristics and record evidence of potentially motile life; while visual changes in coloration may be associated with a non-biological process, most seemingly diel environmental color changes on Earth are associated with biological mechanisms, such as the mats at MIS and bioluminescence by a variety of organisms small and large. The newer type of time-lapse imagery that we suggest, deploying a camera to take many images over a long period of time, covers low spatial area, but high temporal density. This is different than observing entire extraterrestrial bodies change color from far away, or even large visual surveys when on the planetary object. Deployed instruments, such as field mass spectrometers, can also scan for elements and compounds capable of life-support and products of metabolism. However, caution must be exercised in carrying out any such tracer experiments or equipment deployments as they carry considerable risk of contaminating pristine extraterrestrial ecosystems with Earth-based or space vessel-based microbes [42]. The idea of intense monitoring of one location stresses the importance of testing the environment for signs and suitability of life beforehand so than minimal time and resources are wasted.

Life is unlikely to be distributed uniformly or randomly in extraterrestrial hydrospheres—there are likely to be hotspots of life where conditions are, or were, optimal. Thus, the MIS model, with hotspots of biodiversity and activity closest to ground water sources, offers a realistic working model to carry out dry runs (e.g., equipment testing) before doing the real runs in space. The fact that MIS is ice-covered during the winter months and ice-free during the remainder of the year also provides a variable context for making ice-free and under-ice explorations (Figure 6). Although the ice-covered conditions at MIS provide a good testing environment, adapting these ice testing conditions to an extraterrestrial setting will present a significant engineering challenge given the variables of ice depth and composition. Planning for a lander to bore through more than 1–2 m of ice is unrealistic at this point. Another important consideration is the locomotion of the vehicle containing such sampling devices. Historically, space agencies have focused much more on terrestrial and aerial vehicles, but development and deployment of underwater vehicles are well underway, and a variety of means for planetary surface mobility and exploration are now available and could be utilized to study different surface and sub-surface habitats [43].

Similar ROV/AUV operations equipped with landing capability can be carried out on exposed, semi-submerged, and deep-water hard substrates, such as the microbialites of Laguna Bacalar (Figure 7). Telescopic sampling devices could be utilized to sample mats found at greater depths than the vehicle’s reach. Equipment, such as sensor-rich sondes, could be lowered on a winch to even greater depths to characterize the environmental conditions below the surface. Similar to the issue of boring through thick ice coverage is the issue of coring deep potential microbialites (Figure 7B). Aquatic depths beyond 1–2 m depth will present a challenge, that may require a combination of methods of Figure 6 and Figure 7 in order to core deep microbialites.

Mat worlds come in many different forms and are found on or in a variety of substrates. However, the predominant types in extraterrestrial habitats (based on our earthly experience) are likely to be soft mats on soft or hard substrata and mats that occur as a subsurface laminar layer within hard substrata. We propose “suction” as a suitable technique for sampling soft mats found over soft or hard substrata (such as the soft mats found in Lake Huron’s Sinkholes) and “coring” for sampling life embedded within hard substrata (such as the subsurface laminar life found in the Laguna Bacalar microbialites (Figure 8). While simple hydraulics or pumps could be used to suck soft samples up, taking samples from hard materials may be done by combining microspine grips with the integration of coring mechanisms to extract samples from rock in the deep ocean. For example, instruments capable of deep-sea robotics, such as NASA JPL’s Nautilus Gripper, may be suitable for grabbing onto hard substrates for keeping position while carrying out observation/coring/sampling activities [44]. Indeed, NASA and several private aerospace companies (e.g., Stone Aerospace Inc., Austin, TX, USA) are designing and testing versatile instrumented lander platforms and all-terrain vehicles in analog Earth environments, such as ice-covered temperate lakes and polar environments. These platforms/vehicles, such as Stone Aerospace’s Cryobot (“Archimedes”), are specifically designed to go to space as payloads in future missions for deployment to explore, survey, sample, and study specific extraterrestrial habitats, such as Europa with kilometers-thick ice on the surface, and under-ice liquid ocean environments.

Once collected, the samples could be examined on-site for various biogeochemical markers by the usual gas chromatography set-up capable of detecting indicators of life processes, such as trace emissions of methane gas and oxygen, or the presence of organic matter, such as amino acids [45]. While the individual presence of any of these gasses or organic compounds is not proof of life, collectively they can indicate greater promise of life’s occurrence. More importantly, the samples could be examined by tiny portable laser microscopes that can construct 3D images of microbial life and the images beamed back to the orbiting ship or back to Earth [46]. Such space payload-ready microscopes in a box that also include sampling gear and electronics for image storage and transfer capable of conducting microscopy in the field (including tracking the movement of cells) are being developed today and being tested at the Earth’s poles [40,41]. Obtaining images resolving microorganisms and videos recording their movements and changes in coloration would be invaluable proof of life concepts. However, such observations should be followed up by actual sampling of the organisms or other non-invasive methods, such as measurement of metabolites, to confirm the presence of life. Observed movement of perceived “life” particles at the microscopic scale in an aquatic setting would also need to be distinguished from simple Brownian motion of non-biological particles.

## 5. Extant Mat Worlds in Ice and Rock Habitats—Examples from Polar Lakes and Earth’s Deep Biosphere

Antarctica’s permanently frozen subglacial lakes are truly one of Earth’s final frontiers—having been sealed off from the surface for millions of years. Approximately 400 subglacial lakes have been now identified in Antarctica, and sampling of a few lakes suggests that the overlying ice as well as the underlying water are rich in microbes and that the sediment is often overlain by microbial mats [24,47,48,49]. Several studies have measured the metabolism of microbes in these ice-covered lakes and found that a variety of metabolic pathways, ranging from microbial photosynthesis to chemosynthesis, play a key role in the subglacial biogeochemistry of these polar lakes [47,50,51]. Parallels between Antarctica’s subglacial lakes and icecaps of Enceladus and the under-ice ocean of Europa have been noted by others. Microorganisms over-winter in the ice of Antarctic lakes [52], and many have remarked on how the lakes themselves may serve as earthly analog ecosystems in our search for potential life on icy moons, such as Europa, orbiting the outer planets of the solar system [53]. Indeed, several investigators of glacial lakes of Antarctica have suggested that Lake Vostok, with an ice cover of over 3 km and geothermal heating from below, may be a good working analog for the ice-covered oceans of Europa [54,55,56,57].

Earth’s deep biosphere consists of sediments beneath the seafloor and crustal rocks everywhere. Several studies have revealed that the Earth’s living zone extends hundreds of meters if not tens of kilometers beneath the continents and the seafloor [58,59]. In these extreme environments, microbial life goes on slowly but surely, using metabolic processes such as carbon oxidation and methanogenesis [60,61]. That microbial cells thrive in sediments hundreds of meters below the seafloor and in Earth’s crustal rocks for more than a kilometer deep suggests various conceivable scenarios in which life could take a foothold and thrive in similarly harsh or harsher extraterrestrial habitats. Indeed, the tolerance of extremophiles, such as methanogens and halophiles, from terrestrial habitats to low temperatures, desiccation, and salinity fluctuations are often considered reasons why extraterrestrial life we may find on Mars and Europa could resemble those on Earth [62].

## 6. Aphotic Mat Worlds—Example of Deep-Water Sinkholes of Lake Huron

The study of aphotic ecosystems, or those in darkness, presents additional challenges. For example, many varieties of extant microbial mat consortia exist in caves and caverns as well deep-water ecosystems on Earth where no sunlight is available. At ~100 m deep, the Isolated Sinkhole in Lake Huron is one such aphotic ecosystem that is fueled by chemosynthetic microbial mats that thrive on the surface of the sediment, bathed in anoxic and sulfidic ground water (Figure 9) [63]. These and other aphotic mat ecosystems (e.g., inside caves, soils, sediments, ice, rocks, etc.) may serve as analogs for mat worlds that are at the far fringes of the solar system and get little sunlight or ones whose sunlight reception is attenuated by the overlying atmosphere/ice/rocks/aqueous media before reaching the live mat zones. However, such aqueous ecosystems should be amenable for robotic exploration when equipped with a light source. Running practice studies with a variety of extant aphotic ecosystems on Earth will be useful in engineering a versatile toolkit capable of effectively exploring similar or different extraterrestrial habitats receiving no light from their host sun.

## 7. Extant Mats and Microbialites as Testbeds for Study of Extraterrestrial Life

Analogs of the simple benthic microbial mats that prevailed in the early biosphere and dominated life for the first few billion years are still found in refugia all around the Earth, and could be potentially inhabiting extraterrestrial hydrospheres. Furthermore, so much of life in the biosphere is hidden from our senses below the surface and remains understudied in terms of their evolutionary, physiologic, biogeochemical, and biodiversity potential. For example, geothermal light fuels photosynthetic life in the deep sea [65], microbial life thrives beneath the vast Antarctic shelf [66], and life persists in the deep biosphere within sub-seafloor sediments and even the fluids within crustal rocks [60,61].

The fact that abundant and diverse life occurs hidden in subsurface habitats world-wide makes the case for why we should be probing, imaging, and studying extant subsurface life for a more complete understanding of Earth’s biosphere. Furthermore, we should adapt lessons learned here on Earth in our pursuit for signs and signatures of life in extraterrestrial habitats. In this pursuit, extant Earthly microbial mat communities should serve as useful theoretical and practical models in our search for extraterrestrial life in waters such as those of the subglacial oceans of Europa. This is not to suggest, however, that we have mastered the study of extant extreme environments on this planet and that the methods that work here can be adapted seamlessly for exploring even similar-looking extraterrestrial ecosystems. Surprises are sure to crop up in space, and we will have to adapt by going prepared for it with a versatile tool kit of working methodologies. It is not inconceivable that we will encounter familiar or bizarre mat worlds in our ongoing search for life on habitable moons, planets, and exoplanets—perhaps even witnessing, therein, life’s daily movements synchronized to the tempo of their own planetary spins around their host sun, just as it is here on Earth [35,67]. Our adventures working with earthly extreme ecosystems will have prepared us better to design flexible investigative tools and methodologies to better chronicle such life and their activities when we encounter them in space.

Towards the very end of her insightful and witty science book, “The End of Everything”, cosmologist Katie Mack writes that, “Like any explorer reaching the edges of the map, we reach out not knowing what we might find” [68]. In the long run, probing for microbial mat worlds that may be out there may be more rewarding than we can imagine. As Steven Jay Gould observed, “The simplest life may pervade the cosmos” [69], and life’s edge may extend well past our earthly boundaries. Even though we only have evidence of life on Earth so far, “the universe abounds with the chemistry of life” [70]—suggesting that the obvious place to search for extraterrestrial life first is here in our own solar system, where life once formed and may have seeded and reseeded multiple habitable spaces over time [71]. Ongoing and future missions to Mars and the intriguing moons of the outer planets in the solar system offer exciting opportunities for making new discoveries—perhaps finding evidence of past life that is now extinct or even thriving extant extraterrestrial life. Like explorers who have come to the edge of their map, we probe ahead, not knowing what we might find.

## Figures and Tables

**Figure 1 life-11-00883-f001:**
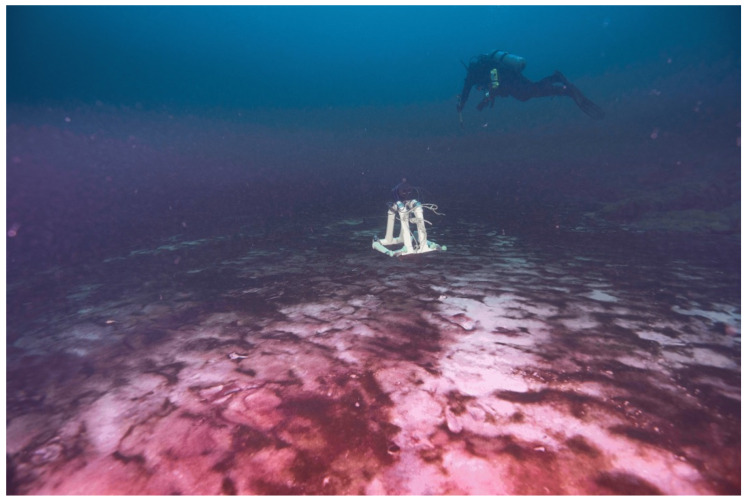
Underwater cameras (GoPro in diver hands and a deployed Marine Imaging Technologies underwater imaging time-lapse camera) capture diel vertical migration in the mat world: microbial mats composed of photosynthetic cyanobacteria (purple patches) and chemosynthetic microbes (white patches) in Middle Island Sinkhole (MIS, at 23 m depth, Lake Huron) take turns at the mat surface during day and night, respectively. Motile mats such as these on the Precambrian seafloor may have laid the foundation for the modern biosphere by optimizing photosynthesis, chemosynthesis, organic carbon burial, and oxygen export. Such modern-day mat ecosystems in submerged sinkholes may also serve as useful working models in our ongoing search for life in extraterrestrial hydrospheres. Scale varies along the photo. For reference, the white camera stand in the background is ~0.5m wide and tall, and the diver above is ~2 m long.

**Figure 2 life-11-00883-f002:**
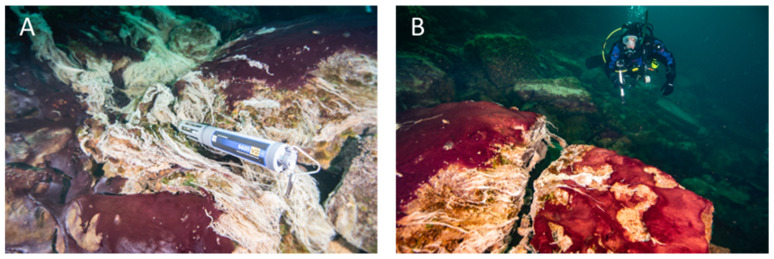
Underwater diver photos showing the predominantly purple-pigmented cyanobacteria along with white patches and strings of chemosynthetic sulfur oxidizing bacteria (note some strings streaming in the low oxygen sulfidic groundwater). A multiparameter water quality sonde (Yellow Springs International) collects time-series environmental data in the groundwater flowing over the mats (**A**), and a diver uses a GoPro camera to capture images of mats covering the limestone ledge over which groundwater is flowing, fueling extensive microbial growth (**B**). Scale varies along the photos. For reference, the sonde is ~0.7 m long (**A**) and the diver is ~0.6 m wide (**B**).

**Figure 3 life-11-00883-f003:**
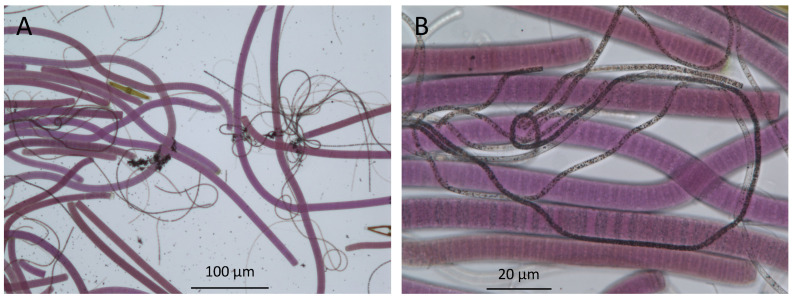
Bright-field microscopy of microbes in the upper millimeters of a mat from Lake Huron’s sinkholes reveal the presence of purple cyanobacteria and filamentous white chemosynthetic bacteria (seen here as dark filaments). (**A**), low magnification 100× image; (**B**), high magnification 400× image).

**Figure 4 life-11-00883-f004:**
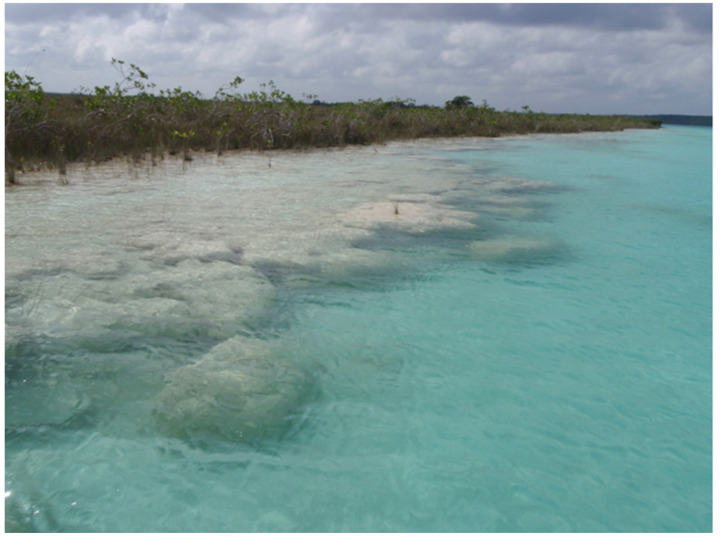
Overview of massive underwater microbialites along the southwestern shoreline of freshwater Laguna Bacalar, Quintana Roo, Mexico. These microbialites or “living rocks” thrive in the shallow carbonate-rich waters of Laguna Bacalar, where a subsurface layer of living cyanobacterial filaments grow their lithified homes daily. Scale varies along the photo. For reference, the microbialites in the foreground are ~3 m wide and in the background are mangrove shrubs ~2–3 m tall.

**Figure 5 life-11-00883-f005:**
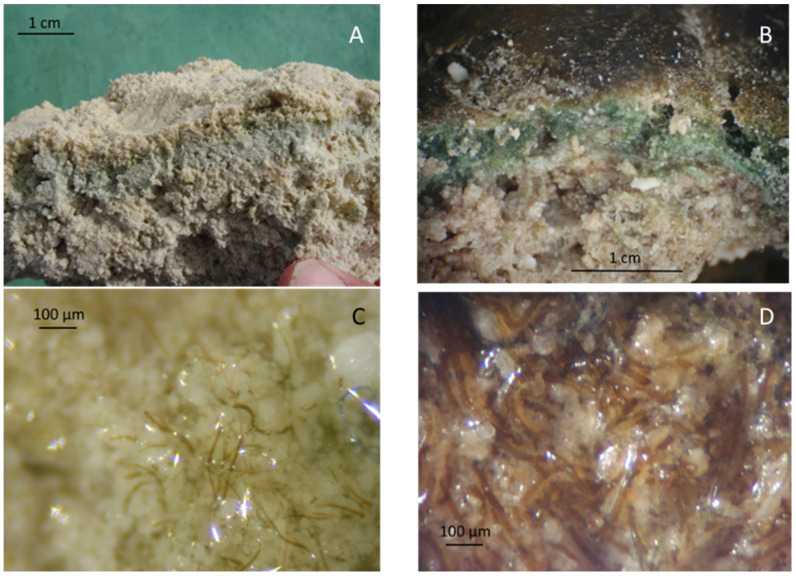
Photos and microscope images of the subsurface living layer in the microbialites of Laguna Bacalar. (**A**) Photo of cross-section of the interior of microbialite showing the horizontal subsurface blue-green pigmented living layer. (**B**) Close-up photo of a cross-section of a microbialite, revealing subsurface blue-green (upper) and purple (lower) pigmented layers (Note: The crusty brown layer on the top is likely mineral; however, it remains unidentified to date (**C**) Microscopic image of the blue-green-colored subsurface zone in B, showing blue-green-pigmented cyanobacterial filaments within. (**D**) Microscopic image of the purple-pigmented layer in B, showing purple cyanobacterial filaments. Methods note: In the field, microbialite sections were made using a hand knife and imaged by an iPhone (**A**,**B**). Microscopy was done using a standard optical dissection scope at 200–400× magnification, and image capture was accomplished by a steady hand holding the lens of the iPhone to the eye piece (**C**,**D**).

**Figure 6 life-11-00883-f006:**
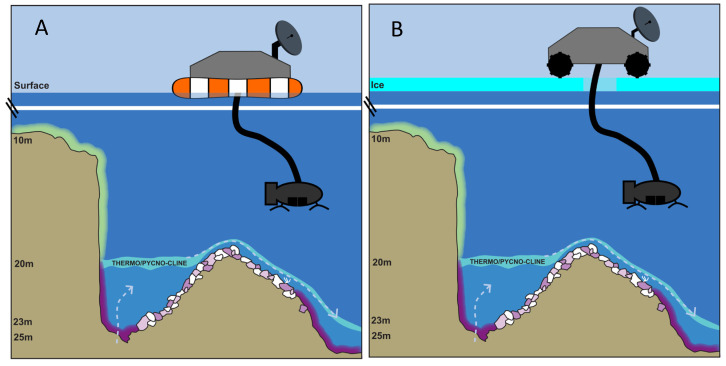
Model scenarios for robotic exploration (video and sampling) of a microbial mat ecosystem on soft sediments as well as hard rock substrata in deep water (but still sunlit), such as in the microbial mat ecosystem in Middle Island Sinkhole, Lake Huron. Here, groundwater enters through the floor of the along-shore sinkhole depression (on the left side of the diagrams) and overflows as an underwater waterfall into the main lake floor, slowing and spreading out as it goes (right side of the diagrams). Scenarios of operations using a floating lander in open water (**A**) and a land rover capable of coring through ice on ice-covered (**B**) mat worlds are depicted using the example of the Middle Island Sinkhole, Lake Huron.

**Figure 7 life-11-00883-f007:**
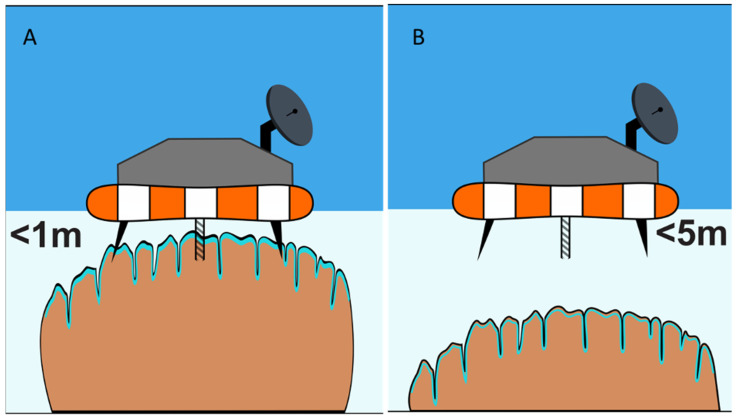
Schematic diagram of a floating lander-type system for studying (video and sensors) and sampling (coring) of shallow water microbial mat systems, such as the microbialites of Laguna Bacalar (**A**). Extensible and retractable coring systems will be needed to sample slightly deeper habitats (**B**).

**Figure 8 life-11-00883-f008:**
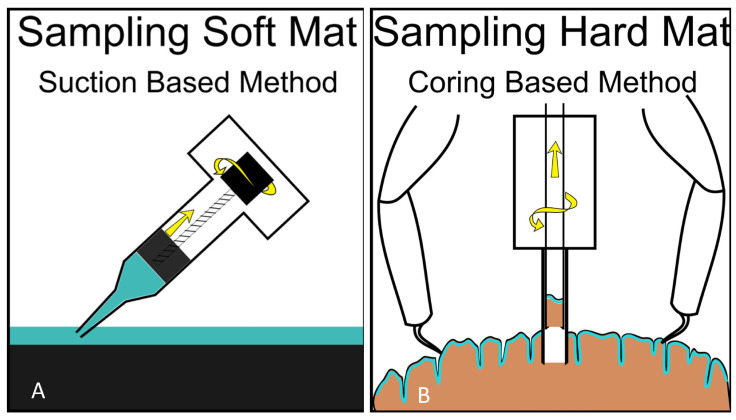
Possible suction-based design (“syringe slurper”—operated by a screw function, as shown, or by piston function) for sampling soft mats, such as the microbial mats of Lake Huron (**A**) and a double-barrel piston corer (with possible screw function for sample extraction and grapples for position-fixing while coring) for sampling microbial life within hard substrates, such as the microbialites of Laguna Bacalar (**B**).

**Figure 9 life-11-00883-f009:**
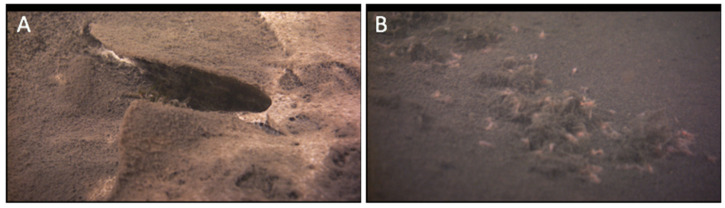
ROV images of the Isolated Sinkhole in Lake Huron, representing a mat world scenario that is under sulfur-rich, anoxic, and aphotic conditions—but one that sustains life in the form of a network of chemosynthetic filaments spread out on the sediment surface in a brown and white layer at a depth of 110 m (**A**); for scale, the width of the U-shaped depression from which the groundwater is actively venting is ~10 cm; see Biddanda et al. 2006 for more information). Here, in an always-dark world, chemosynthetic microbes, such as large sulfur-oxidizing microbes (large pinkish rod-shaped cells in the image), are the primary producers (**B**); for scale, the large pinkish cells are ~0.5 mm; see [64] for more information).

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
