# Peer review of "Extant Earthly Microbial Mats and Microbialites as Models for Exploration of Life in Extraterrestrial Mat Worlds"

_life, 2021, doi:10.3390/life11090883_

Round 1
Reviewer 1 Report
Overview:
This paper reviews the work done by the authors on a soft microbial mat and a hard microbialite, as they show that each type of microbial environment is representative of more of its kind on Earth. Then, using these two microbial environments as models, they review sampling techniques that have been applied in order to suggest methods for potential recovery of samples from similar environments that might house life on planets and moons in our solar system.
General comments:
The authors have a good command of the English language, however, the overuse of quotes, exposition, and flowery language – in particular in the introduction and conclusion – detract from the science presented. While the use of these types of writing do make this paper easy to read, I suggest that the authors heavily revise the manuscript.
The main purpose of this manuscript is to illustrate how sampling techniques, used on extant mats and microbialites, could be added to future space exploration missions as a way to capture life. The authors, however, fail to demonstrate that the Lake Huron sinkhole mat and the Laguna Bacalar microbialite are representative of other known mats and microbialites, respectively. Given that this is a review paper, there should be at least a paragraph for each environment outlining why these environments are representative and, thus, usable models for sampling techniques. The authors should at least compare the work done at their environments to the work done at Little Salt Spring, Florida, or the tufa’s of Big Soda Lake, Nevada – these are just two examples.
Ultimately, this review does not provide a sufficient overview of mats and microbialites to allow it to fulfill its title and the small amount of well thought out real-world science and its subsequent discussion are hidden by excess descriptive language.
Line by line comments:
Line 40-41: add reference to Whitman 1998 “The unseen majority”
Line 148: remove “nutrient poor waters of” – it is sufficient to say “oligotrophic Lake Bacalar”
Line 155-156: unnecessary text – simply say “…beneath the limestone surface, likely to prevent UV-photodamage.”
Line 175: This is a stylistic choice, however, I would suggest rewriting the “Note” in more formal terms, such as “the crusty brown layer is likely mineral, however, it remains unidentified to date.”
Line 277-278: remove the end of the sentence, it is redundant with the beginning. “using a “suction” and “corer” type systems, respectively.”
Line 286: change “in ice-covered” to “such as ice-covered”
Line 297: remove “the usual”
Author Response
Response to Reviewer Feedback:
We are thankful to the three reviewers for constructive comments and suggestions for improving our manuscript. We have carefully considered each of the Reviewer concerns and addressed them to the best of our ability, and hope that the revised manuscript will be acceptable to Life.
Below, we give an item-by item response with reviewer comments in italics and our response in regular font, giving the location numbers of the changes made in each case inside parenthesis ().
Reviewer 1 Comment 1:
This paper reviews the work done by the authors on a soft microbial mat and a hard microbialite, as they show that each type of microbial environment is representative of more of its kind on Earth. Then, using these two microbial environments as models, they review sampling techniques that have been applied in order to suggest methods for potential recovery of samples from similar environments that might house life on planets and moons in our solar system. The main purpose of this manuscript is to illustrate how sampling techniques, used on extant mats and microbialites, could be added to future space exploration missions as a way to capture life.
General comments: The authors have a good command of the English language, however, the overuse of quotes, exposition, and flowery language – in particular in the introduction and conclusion – detract from the science presented. While the use of these types of writing do make this paper easy to read, I suggest that the authors heavily revise the manuscript.
Author Response: We have now had the paper read by a colleague with better command of the English language which affected numerous corrections that address the Reviewer concern. We have confined the use of quotes to the very last paragraph and minimized the use of flowery language throughout the manuscript.
Reviewer 1 Comment 2: The authors, however, fail to demonstrate that the Lake Huron sinkhole mat and the Laguna Bacalar microbialite are representative of other known mats and microbialites, respectively. Given that this is a review paper, there should be at least a paragraph for each environment outlining why these environments are representative and, thus, usable models for sampling techniques. The authors should at least compare the work done at their environments to the work done at Little Salt Spring, Florida, or the tufa’s of Big Soda Lake, Nevada – these are just two examples. Ultimately, this review does not provide a sufficient overview of mats and microbialites to allow it to fulfill its title and the small amount of well thought out real-world science and its subsequent discussion are hidden by excess descriptive language.
Author Response: We now paint a better context of the globally distributed mat and microbialite life as 2 distinctive types of extant microbial life, and their usefulness as models for exploration of extraterrestrial life (lines 2-54; 117-123) .
We have made numerous edits to further address the reviewers point about why the Lake Huron’s mats are representative of globally distributed “soft mats”, and Laguna Bacalar microbialites as representative of “hard mats” (117-120; 159-161).
Reviewer 1 Comment 3: Line by line comments: Line 40-41: add reference to Whitman 1998 “The unseen majority” Line 148: remove “nutrient poor waters of” – it is sufficient to say “oligotrophic Lake Bacalar” Line 155-156: unnecessary text – simply say “…beneath the limestone surface, likely to prevent UV-photodamage.” Line 175: This is a stylistic choice, however, I would suggest rewriting the “Note” in more formal terms, such as “the crusty brown layer is likely mineral, however, it remains unidentified to date.” Line 277-278: remove the end of the sentence, it is redundant with the beginning. “using a “suction” and “corer” type systems, respectively.” Line 286: change “in ice-covered” to “such as ice-covered”
Author Response: Whitman et al. 1998 is now correctly cited in the 1st paragraph (line 38). In a similar vein, we have agreed with and addressed all the remaining suggestions on oligotrophy (156), UV-photodamage (165); Note (183-184); suction and corer (307-309), and ice-covered (317).

Reviewer 2 Report
The purpose with this study is to study terrestrial microbial mat systems and evaluate how the methods used for identifying this microbial life systems could be used in extraterrestrial worlds. The authors have investigated two systems, microbial mats in Lake Huron and the Laguna Bacalar microbialites. It is an interesting read but it would need some more work. Nothing is said about any problems or difficulties of any of the suggested methods and all microbial work on the sites have already been done by others. Are there any ambiguities of difficulties with the suggested methods, and if so, what can be done to overcome these ambiguities?What are the risks of contamination? What is a solid evidence for life? Much of these methods presented here have already been suggested as methods or techniques for space travels. What is new with this study? I believe the authors need to expand the discussion quite much and to clearly state what is new and unique with their study.
Line 108-110
Why is this presumed? Please add some more details on this, including references.
Line 191
Re-phrase. There are many bacteria that have typical morphological features similar to Calothrix and Scytonema, so I would say it is impossible to make such an observation without PCR.
Line 231
Even if it is possible to inject dyes in terrestrial environments due to our excessive abundance of life, I would find it quite unlikely that it would be an option in extraterrestrial worlds, due to the risk of contamination of the environment.
Line 229 and 233
No non-biological changes in morphology and color is discussed at all. Are there any non-biological features that are similar to those observed in this study? If so, how can we differentiate between these two structures? Add some more discussion about ambiguities and difficulties and how to overcome those.
Line 296-306
I agree that methane, oxygen and amino acids are nice indicators for life but I still think this should be more discussed, since none of these are unambiguous evidences for life and many non-biological processes can form all of these gases and organic compounds. Tracking movement of cells is however a quite good idea but should also be backed up with additional evidence that it is actually life that is recorded.
Author Response
Response to Reviewer Feedback:
We are thankful to the three reviewers for constructive comments and suggestions for improving our manuscript. We have carefully considered each of the Reviewer concerns and addressed them to the best of our ability, and hope that the revised manuscript will be acceptable to Life.
Below, we give an item-by item response with reviewer comments in italics and our response in regular font, giving the location numbers of the changes made in each case inside parenthesis ().
Reviewer 2 Comment 1: The purpose with this study is to study terrestrial microbial mat systems and evaluate how the methods used for identifying this microbial life systems could be used in extraterrestrial worlds. The authors have investigated two systems, microbial mats in Lake Huron and the Laguna Bacalar microbialites. It is an interesting read but it would need some more work. Nothing is said about any problems or difficulties of any of the suggested methods and all microbial work on the sites have already been done by others. Are there any ambiguities of difficulties with the suggested methods, and if so, what can be done to overcome these ambiguities? What are the risks of contamination? What is a solid evidence for life? Much of these methods presented here have already been suggested as methods or techniques for space travels. What is new with this study? I believe the authors need to expand the discussion quite much and to clearly state what is new and unique with their study.
Author Response: In the revised manuscript, we have addressed Reviewer concerns by pointing out advantages as well as difficulties of working with extant microbial mat and microbialite ecosystems as models for exploration of mat worlds in space (244-259; 292-295; 309-311; 328-344; 410-425), risks of contamination of extraterrestrial environments by the proposed methods of study (254-257) and expanded the discussion by pointing out what is new in this study – for example, the use of time-lapse photography that does not require sampling (244-259) and other non-invasive methodologies (328-344). We have refrained from addressing the question of “What is a solid evidence life in space?” – as it is a huge topic in itself and any answer is likely to be subject to further discussion. We do, however, discuss to some extent why more than one biomarker of life would be useful and that non-biological particles should not be confused with biological material (328-344).
Reviewer 2 Comment 2: Line 108-110 Why is this presumed? Please add some more details on this, including references.
Author Response: We now explain the basis for this presumption – such as the measured daily variability in pH in the mat-sediment complex and cite the literature for data from the environment (108-114).
Reviewer 2 Comment 3: Line 191 Re-phrase. There are many bacteria that have typical morphological features similar to Calothrix and Scytonema, so I would say it is impossible to make such an observation without PCR. Reading
Author Response: We agree with this point, and have made it clear that independent confirmation will require additional studies (198-202).
Reviewer 2 Comment 4: Line 231 Even if it is possible to inject dyes in terrestrial environments due to our excessive abundance of life, I would find it quite unlikely that it would be an option in extraterrestrial worlds, due to the risk of contamination of the environment.
Author Response: We agree with this point, and have no ready solution. We do express the need for exercising caution in all types of extraterrestrial experiments and deployments) that carry risks of contamination (254-257).
Reviewer 2 Comment 5: Line 229 and 233 No non-biological changes in morphology and color is discussed at all. Are there any non-biological features that are similar to those observed in this study? If so, how can we differentiate between these two structures? Add some more discussion about ambiguities and difficulties and how to overcome those.
Author Response: We now discuss the need for distinguishing between biological and non-biologically similar structures (340-344).
Reviewer 2 Comment 6: Line 296-306 I agree that methane, oxygen and amino acids are nice indicators for life but I still think this should be more discussed, since none of these are unambiguous evidences for life and many non-biological processes can form all of these gases and organic compounds. Tracking movement of cells is however a quite good idea but should also be backed up with additional evidence that it is actually life that is recorded.
Author Response: We fully agree with the Reviewer’s comment. While we do not have a fool-proof working solution to propose, we do discuss the value of making multiple biomarker measurements and the need of distinguishing between biological and non-biological materials (244-259).
Reviewer 3 Report
The authors of the articles “Extant earthly microbial mats and microbialites as models for exploration of life in extraterrestrial mat worlds” describes the studies of two distinct Earthly microbial mat ecosystems and ponder how similar or modified methods of study (e.g., robotics) would be applicable to prospective mat worlds in other planets and their moons (e.g., subsurface Mars, and under-ice oceans of Europa). The manuscript is well written and provide a review of the new prospective of Mat colonization.
The Manuscript should be acceptable after minor revision:
Line 87 “compared to overlying Lake Huron water” please specify the location of the Lake Huron
Line 183 please add the place i.e. Mexico in the “Laguna Bacalar”
Line 201-205 please reprase this sentence why
Line 236 changes “hydospheres” in “hydrospheres”
Line 238 please could you explain better this point ? “offers a realistic working model to carry out 238 dry runs before the real runs in space”
Line 54 Maybe the authors could cite the recent studies of shallow hydrothermal vents as extraterrestrial analogs Zammuto et al 2018 and/or Zammuto et al 2020.
Author Response
Response to Reviewer Feedback:
We are thankful to the three reviewers for constructive comments and suggestions for improving our manuscript. We have carefully considered each of the Reviewer concerns and addressed them to the best of our ability, and hope that the revised manuscript will be acceptable to Life.
Below, we give an item-by item response with reviewer comments in italics and our response in regular font, giving the location numbers of the changes made in each case inside parenthesis ().
Reviewer 3 Comment 1: Comments and Suggestions for Authors The authors of the articles “Extant earthly microbial mats and microbialites as models for exploration of life in extraterrestrial mat worlds” describes the studies of two distinct Earthly microbial mat ecosystems and ponder how similar or modified methods of study (e.g., robotics) would be applicable to prospective mat worlds in other planets and their moons (e.g., subsurface Mars, and under-ice oceans of Europa). The manuscript is well written and provide a review of the new prospective of Mat colonization. The Manuscript should be acceptable after minor revision:
Line 87 “compared to overlying Lake Huron water” please specify the location of the Lake Huron Line 183 please add the place i.e. Mexico in the “Laguna Bacalar” Line 201-205 please reprase this sentence why Line 236 changes “hydospheres” in “hydrospheres”
Author Response: We have now added the location of the 2 study sites (63-70), and corrected the spelling error associated with “hydrospheres” (260-261).
Reviewer 3 Comment 2: Line 238 please could you explain better this point ? “offers a realistic working model to carry out 238 dry runs before the real runs in space” Reading
Author Response: Thank you for pointing out this error. We have now corrected and clarified this sentence “Thus, the MIS model, with hotspots of biodiversity and activity closest to ground water sources, offers a realistic working model to carry out dry runs (e.g., equipment testing) before doing the real runs in space.” (262-295).
Reviewer 3 Comment 3: Line 54 Maybe the authors could cite the recent studies of shallow hydrothermal vents as extraterrestrial analogs Zammuto et al 2018 and/or Zammuto et al 2020.
Author Response: Thank you for pointing out this important literature that we had missed. We have now cited both papers in the relevant parts of the manuscript (52 and 257).
Round 2
Reviewer 2 Report
The authors have made a good and thorough job with the review and I have no further questions or comments to the manuscript.